# A Method for Detecting the Vacuum Degree of Vacuum Glass Based on Digital Holography

**DOI:** 10.3390/s23052468

**Published:** 2023-02-23

**Authors:** Ting Li, Qinghe Song, Guangjun He, Haiting Xia, Haoxiang Li, Jinbin Gui, Haining Dang

**Affiliations:** 1Faculty of Science, Kunming University of Science and Technology, Kunming 650500, China; 2Yunnan Key Laboratory of Disaster Reduction in Civil Engineering, Kunming 650500, China; 3Faculty of Civil Engineering and Mechanics, Kunming University of Science and Technology, Kunming 650500, China

**Keywords:** digital holography, optical pressure sensor, vacuum degree, vacuum glass

## Abstract

The vacuum degree is the key parameter reflecting the quality and performance of vacuum glass. This investigation proposed a novel method, based on digital holography, to detect the vacuum degree of vacuum glass. The detection system was composed of an optical pressure sensor, a Mach–Zehnder interferometer and software. The results showed that the deformation of monocrystalline silicon film in an optical pressure sensor could respond to the attenuation of the vacuum degree of vacuum glass. Using 239 groups of experimental data, pressure differences were shown to have a good linear relationship with the optical pressure sensor’s deformations; pressure differences were linearly fitted to obtain the numerical relationship between pressure difference and deformation and to calculate the vacuum degree of the vacuum glass. Measuring the vacuum degree of vacuum glass under three different conditions proved that the digital holographic detection system could measure the vacuum degree of vacuum glass quickly and accurately. The optical pressure sensor’s deformation measuring range was less than 4.5 μm, the measuring range of the corresponding pressure difference was less than 2600 pa, and the measuring accuracy’s order of magnitude was 10 pa. This method has potential market applications.

## 1. Introduction

Vacuum glass is a new type of deep processing glass product; its vacuum degree in two glass cavities is almost a vacuum [1,2]. Vacuum glass has the advantages of low-carbon energy saving [3,4], sound insulation, noise reduction, heat insulation and heat preservation [5,6], and is widely used in construction, transportation, photovoltaic integration and so on [5,7,8]. For example, in 2017, China’s demand for vacuum glass was 777,000 square meters, with a market scale of approximately CNY 747 million; by 2024, China’s vacuum glass production is expected to reach 2.449 million square meters, with a market scale of approximately CNY 2.436 billion. Vacuum degree is the key parameter reflecting the quality and performance of vacuum glass. Given vacuum glass’ wide applications and popularity, research regarding vacuum degree measurement has become urgent.

At present, measurement of the vacuum degree of vacuum glass is sensorless, using either photoelastic or thermal conductivity detection. The photoelastic method uses the relationship between the stress spot of the vacuum glass’ support and pressure to obtain the vacuum degree. Liu et al. used this method semi-quantitatively to measure the vacuum degree of vacuum glass [9]. The thermal conduction method characterizes the vacuum degree using the temperature change rate on both sides of the vacuum glass. Li et al. employed the thermal conduction method to obtain the vacuum degree of vacuum glass via infrared heating. The problem with these methods is that their vacuum degree detection accuracy is not high enough. In addition, although the sensor method can detect the vacuum degree [10,11,12,13], it cannot detect the vacuum degree of vacuum glass, because the sensor method’s circuit feed requires more space than is available [10]. To sum up, it is necessary to conduct in-depth research regarding the detection of the vacuum degree of vacuum glass [14].

Digital holography is an advanced optical testing method, which has the advantages of being nondestructive, high precision, noncontact, etc. [15]. Digital holography has been widely used in three-dimensional morphology detection [16,17,18], temperature field detection [19,20], micro vibration [21], etc. Therefore, digital holography has potential application prospects in detecting the vacuum degree of vacuum glass. However, there are no relevant reports regarding the application of digital holography to the measurement of the vacuum degree of vacuum glass.

In this study, a novel method to detect the vacuum degree of vacuum glass was proposed, based on digital holography. An optical path of digital holographic interferometry, based on the Mach–Zehnder interferometer, was established; a finely fabricated optical pressure sensor was the system’s core component. Recording and reconstructing the hologram, we obtained the numerical relationship between the pressure difference and the optical pressure sensor’s deformation. Finally, the vacuum degrees of vacuum glass under three air leakage levels were accurately measured using this system, which was expected to rapidly and accurately detect the vacuum degree of vacuum glass.

## 2. Principle and Methods 

### 2.1. Principle of Digital Holography

Digital holography uses CCD to record the interference between an object light wave U(x,y) and reference light wave R(x,y); the recording distance is $z_{0}$, as shown in Figure 1. The intensity of the interference field I(x,y) can be expressed as:(1)$$I\left(x,y\right)={\left|U\left(x,y\right)\right|}^{2}+{\left|R\left(x,y\right)\right|}^{2}+R\left(x,y\right)U^{*}\left(x,y\right)+R^{*}\left(x,y\right)U\left(x,y\right)$$
where $R\left(x,y\right)U^{*}\left(x,y\right)$ carries valuable phase information regarding the object. The conjugate light $R\left(x,y\right)U^{*}\left(x,y\right)$ can be reconstruction using the FIMG4FFT reconstruction algorithm [22,23]. The reconstructed image, with none-zero diffraction order, was obtained using the window filtering function H(x,y). The reconstructed object field at the reconstruction distance $z_{i}$ under spherical wave illumination can be expressed as:(2)$$U\left(x,y\right)=F^{-1}\left\{F\left\{R\left(x,y\right)U^{*}\left(x,y\right)H\left(x,y\right)R_{c}\left(x,y\right)\right\}\times \exp\left[jkz_{i}\sqrt{1-\lambda^{2}\left(f_{x}^{2}+f_{y}^{2}\right)}\right]\right\}$$
where $R_{c}\left(x,y\right)$ is a reconstructed spherical wave, $f_{x}$ and $f_{y}$ are the sampling interval in the frequency domain, λ is the light wavelength, F represents the fast Fourier transform and $F^{-1}$ represents the inverse fast Fourier transform.

The phase can be expressed as follows:(3)φ(x,y)=arctan[ImU(x,y)ReU(x,y)]

The phase φ(x,y) is generally wrapped between (−π,π); the unwrapped phase can be obtained using the iterative least square method [24].

The object’s deformation causes a change in optical path difference, resulting in a change in phase difference, as shown in Equation (4):(4)$$\Delta \varphi \left(x,y\right)=\varphi_{2}\left(x,y\right)-\varphi_{1}\left(x,y\right)=\frac{2\pi }{\lambda }w\left(x,y\right)$$
where w(x,y) is the deformation of the object.

### 2.2. Digital Holographic Optical Path

Based on the basic theory of digital holography, we designed an interferometric optical path Mach–Zehnder interferometer to measure the vacuum degree of vacuum glass. The digital holographic optical path is shown in Figure 2, and operates as follows. The light emitted by the laser becomes a parallel light after being collimated using a microscopic objective (MO), a pinhole filter (PF) and a Fourier lens (FL). Next, the parallel beam is divided into reference light and measurement light using a nonpolarizing beam splitter (NBPS1). The measurement light propagates down to the optical pressure sensor inside the vacuum glass to form scattered light, while the reference light, reflected by a plane mirror, propagates to the right and downward, followed by the beam combination with scattered light in the NPBS2. Finally, the interference hologram is recorded using a charge coupled device (CCD) and reconstructed by the computer.

### 2.3. Optical Pressure Sensor

#### 2.3.1. Basic Theory

According to the theory of elasticity, the film in the optical pressure sensor we designed can be equivalent to a thin plate [25,26,27]. The differential equation for the small-deflection elastic surface of film is:(5)$$D\nabla^{4}w=P-P_{1}$$
where P is the pressure on one side of the film, $P_{1}$ is the pressure on the other side of the film and D is the bending stiffness of the thin plate.

For films with radius r=a. and fixed circumference, the boundary condition is:(6)$$\left\{ \begin{array}{c} w(r){|}_{r=a}=0\\ \frac{dw\left(r\right)}{dr}{|}_{r=a}=0 \end{array}\right.$$

The maximum deformation of the circular film is located at its center (r=0), and represented as follows:(7)$$w{\left(r\right)}_{\max}=\frac{\left(P-P_{1}\right)a^{4}}{64D}=\frac{\left(P-P_{1}\right)a^{4}\left(1-\mu^{2}\right)}{16Eh^{3}}$$
(8)$$\Delta w=w{\left(r\right)}_{\max1}-w{(r)\prime }_{\max1}=\frac{({P\prime }_{1}-P_{1})a^{4}\left(1-\mu^{2}\right)}{16Eh^{3}}$$
where E is the elastic modulus, μ is Poisson’s ratio and h is the film’s thickness.

Equation (8) theoretically gives the numerical relationship between the deformation and the circular film’s pressure difference in the optical pressure sensor.

#### 2.3.2. Preparation Method

The optical pressure sensor was fixed inside the vacuum glass, as shown in Figure 3b, and was composed of a base and a film, as shown in Figure 3a. The base was a cylindrical cavity with an opening at one end, made of glass. The film was tightly adhered to the base’s opening to form a tiny cavity with excellent air tightness. The pressure *P* of this tiny cavity was always constant during the detection process, and its value could be given according to the detection range. In this experiment, the pressure of the tiny cavity was standard atmospheric pressure. When the pressure *P*_1_ in the glass cavity changed, the film deformed due to the pressure difference.

Together, the film’s sensitivity, linearity and output characteristics played a decisive role in the optical pressure sensor. Monocrystalline silicon film with a high strain sensitivity coefficient and stable chemical properties was selected for use in the optical pressure sensor. This film has the advantages of large-scale and low-cost processing, and is the best choice for various pressure sensors.

Digital holography’s detection accuracy is at the sub-wavelength level. This optical detection system’s expected accuracy was at the 10 pa level. The monocrystalline silicon film’s geometric parameters were calculated by combining its material parameters, as shown in Table 1.

The vacuum degree detection process for vacuum glass was as follows. First, we pumped the glass cavity using the built-in optical pressure sensor into a certain vacuum; next, we controlled the air valve according to the specified procedure to make it leak slowly. The optical pressure sensor’s holograms before and after air leakage under a certain vacuum degree were recorded. Next, the phase difference was obtained using computer numerical calculation, and the deformation of the film in the optical pressure sensor was obtained based on Equation (4). At the same time, the vacuum degree of the vacuum glass was read using a vacuum gauge (Sanliang DP380, measuring range ±100 Kpa, accuracy 0.01 Kpa) in real time, which allowed the numerical relationship between the pressure difference and the deformation to be established. This operation was repeated many times, and finally, the statistical relationship between the optical pressure sensor’s deformations and the pressure differences could be calibrated.

During the actual measurement process, the film’s deformation in the optical pressure sensor under the change in vacuum degree was obtained using the optical detection system. The vacuum degree could be obtained according to the statistical relationship noted above. All measurement processes were carried out at room temperature.

## 3. Results and Discussion

### 3.1. Deformation Measurement of Monocrystalline Silicon Film in the Optical Pressure Sensor

The vacuum glass, with a built-in optical pressure sensor, was pumped to a certain vacuum degree $p_{0}$ and then placed in the optical detection path. The hologram under this vacuum degree was recorded using CCD, as shown in Figure 4a. The vacuum glass slowly leaked 1070 pa, via control of the air valve; the hologram after the leak was also recorded using CCD, as shown in Figure 4b. Figure 4a,b were interference patterns of object light and reference light, respectively, which carried the strength and phase information for the monocrystalline silicon film in the optical pressure sensor [28,29]. However, changes in strength and phase could not be directly seen from these two images; the hologram required further processing.

The phase information in the hologram was more important for reflecting the three-dimensional information regarding the measured object’s surface [30]. First, using the FIMG4FFT algorithm, the wavefront phase distributions of single crystal silicon film before and after air leakage were numerically reconstructed from the recorded digital holograms. Next, the phase difference was calculated by substituting the phase in these two states into Equation (4). A series of concentric circular stripes is visible in Figure 5a; these stripes were caused by the corresponding out-of-plane displacement of the monocrystalline silicon film in the optical pressure sensor. The distance between stripes reflected the change in phase information with the vacuum degree.

A large amount of speckle noise in the phase difference map affected the efficiency and accuracy of phase unwrapping [31]. Next, the phase difference was filtered using window Fourier transform to reduce the noise and improve the signal-to-noise ratio; the fringe contrast of the phase difference after noise reduction was better, and the fringe contour was clearer. The wrapped phase difference map after noise reduction is shown in Figure 5b.

The wrapped phase difference was unwrapped to obtain its true phase difference distribution because the arctangent function was used in phase processing, and the resulting phase difference was wrapped between [−π, π]. The unwrapped phase difference could be obtained using the iterative least square method, as shown in Figure 5c; the true value of the phase difference after unwrapping was [−30, 0]. Figure 5d is a three-dimensional map of the unwrapped phase difference, which indirectly reflects the changes in the film morphology characteristics with the vacuum degree.

Film deformation could be calculated by substituting the real phase difference into Equation (4), as shown in Figure 6. The three-dimensional film deformation map is shown in Figure 6a; the maximum film deformation was located at the center of the film, which was consistent with results derived from the previous theory. Figure 5b is the middle profile of Figure 6a; when the air leakage was 1070 pa, the deformation at the center of the film was 1.83 μm. This result indicates that digital holography can be used to detect deformation at the micron level, and also proves that it is feasible to detect the vacuum degree of vacuum glass using digital holography.

### 3.2. Variation in Vacuum Degree with Optical Pressure Sensor Deformation

Within the monocrystalline silicon film’s small deflection deformation range [32], we pumped the vacuum glass into different initial vacuums $P_{1}$, measured the film’s deformation under different vacuum attenuations and recorded the corresponding pressure differences. We adopted the cross-subtraction method of pressure difference to eliminate the deviation caused by different initial morphology of the film and took the mean value of multiple deformations at around the maximum deformation as the final measured value. In total, 239 data points of pressure differences and deformations were obtained. Figure 6a illustrates that the monocrystalline silicon film’s deformation changed within (0, 4.5) μm, and the corresponding pressure difference changed within [0, 2600] pa. The smaller the deformation, the smaller the pressure difference and the more concentrated the data points; the pressure difference showed a good linear relationship with the deformation. With an increase in deformation, data point concentration decreased, but there was still a good linear relationship between deformation and pressure difference. These results indicate that the optical pressure sensor could convert the deformation into pressure difference accurately and linearly. In addition, the initial vacuum degree’s influence on the change in the film’s pressure difference with deformation was negligible. When vacuum attenuation exceeded 2600 pa, the fringes in the phase difference were too dense, resulting in the phase information being masked by speckle noise and difficult to reconstruct. Therefore, it was beyond the sensor’s pressure difference measurement range. Of course, if the film’s geometric or material parameters in the sensor were changed, the sensor’s pressure difference measurement range would also change accordingly.

To further verify the change trend of the optical pressure sensor in theory, a curve of pressure difference (ΔP) with deformation (Δw) was plotted, as shown in Figure 7b. Obviously, the pressure difference in the optical pressure sensor was linear with the deformation, which was consistent with the experimental results. Therefore, the variations in pressure difference with deformation in the optical pressure sensor could be fitted using a linear relationship. On the other hand, after Equation (8) was deformed using $\Delta P/\Delta w=\frac{16Eh^{3}}{3a^{4}(1-\mu^{2})}$, the pressure difference in the unit deformation mainly depended on the thickness and radius of the film. It can be inferred that the optical pressure sensor’s measurement range could be improved by increasing the thickness or decreasing the radius.

The fitting formula could be obtained using linear fitting for data points (in Figure 7a) as follows: ΔP=585.65Δw+1.64, $R^{2}=0.99$. Figure 7c presents a comparison between the theoretical and experimental fitting values of pressure difference with deformation; the intercept and slope of the two straight lines are almost equal, and the average error between them is only 2.43% through theoretical calculation. It can be concluded that the optical pressure sensor based on digital holography can be used to measure the vacuum degree of vacuum glass.

Finally, the vacuum degree of vacuum glass can be expressed as: (9)$${P\prime }_{1}=P_{1}+\Delta P$$

### 3.3. Application of Vacuum Degree Measurement of Vacuum Glass

The optical pressure sensor based on digital holography was used to detect the vacuum degree of vacuum glass under different initial vacuum degrees.

When the vacuum glass’ initial vacuum degree was −5 Kpa, the air valve was controlled to let it leak slowly for 2 min; then, the film’s deformations in the optical pressure sensor before and after air leakage were measured. These measurements were repeated for 10 groups. Figure 8a shows the wrapped phase difference map after noise reduction, Figure 8b shows the unwrapped phase difference map, Figure 8c shows the three-dimensional deformation map of film in the optical pressure sensor, and Figure 8d shows the deformation measured for 10 groups. The deformations of these 10 groups were almost equal, and the standard deviation of 0.01672 μm indicated that the fluctuation between these 10 data groups was quite small, which further proves that the detection system’s stability was very high. According to the fitting results, the average value was 160 pa of pressure difference corresponding to 10 deformation groups for the optical pressure sensor; therefore, the vacuum degree of the vacuum glass after leakage was −4.84 Kpa. In comparison, the vacuum degree of the vacuum glass after leakage, measured using an electronic differential pressure meter, was also −4.84 Kpa.

When the vacuum glass’ initial vacuum degree was −10 Kpa, the air valve was controlled to let it leak slowly for 8 min. Figure 9a–c show the wrapped phase difference map after noise reduction, the unwrapped phase difference map and the three-dimensional deformation map of film in the optical pressure sensor, respectively. Figure 9d shows that the deformations of these 10 groups differed only slightly; its standard deviation of 0.02476 μm shows that the fluctuation between these 10 data groups was small. The average value was 500 pa of pressure difference corresponding to 10 deformation groups; therefore, the vacuum degree of the vacuum glass after leakage was −9.50 Kpa. In comparison, the vacuum degree of the vacuum glass after leakage measured using an electronic differential pressure meter was −9.49 Kpa.

When the vacuum glass’ initial vacuum degree was −15 Kpa, the air valve was controlled to let it leak slowly for 13 min. Figure 10a–c show the wrapped phase difference map after noise reduction, the unwrapped phase difference map and the three-dimensional deformation map of film in the optical pressure sensor, respectively. Figure 10d shows that the deformations of these 10 groups differed only slightly, and the standard deviation of 0.10572 μm shows that the fluctuation between these 10 data groups was small. The average value was 1050 pa of pressure difference corresponding to 10 deformation groups; therefore, the vacuum degree of the vacuum glass after leakage was −13.95 Kpa. In comparison, the vacuum degree of the vacuum glass after leakage measured using an electronic differential pressure meter was −14.00 Kpa.

In summary, the digital holographic detection system we built could accurately measure the vacuum degree of vacuum glass, and had the advantages of high precision, fast real-time good stability, and no circuit feed through. Additionally, the miniaturization of the optical pressure sensor was the main challenge in realizing market applications of this technology [33].

## 4. Conclusions

In conclusion, we designed a system to measure the vacuum degree of vacuum glass based on digital holography, composed of an optical pressure sensor, a Mach–Zehnder interferometer and software. The geometric parameters of single crystal silicon film were reasonably selected according to the measurement range and accuracy of vacuum degree; next, the optical pressure sensor was fabricated. Results showed that the digital holographic detection system could detect changes in the monocrystalline silicon film’s deformation using the vacuum degree when the vacuum glass’ vacuum degree decayed. Under different initial vacuum degrees, 239 experimental data groups with different air leaks showed that the pressure difference had a good linear relationship with the optical pressure sensor’s deformation. Based on theoretical derivation, these data were linearly fitted to obtain the numerical relationship between pressure difference and deformation; finally, the vacuum degree of the vacuum glass was calculated. Measuring the vacuum degree of vacuum glass under three different conditions proved that the digital holographic detection system could measure the vacuum degree of vacuum glass quickly and accurately. The optical pressure sensor’s deformation measurement range was less than 4.5 μm, the measurement range of the corresponding pressure difference was less than 2600 pa and the measuring accuracy’s order of magnitude was 10 pa. In addition, by changing the optical pressure sensor’s film parameters, the measurement range or vacuum degree detection accuracy could be improved. This technology has market application potential.

## Figures and Tables

**Figure 1 sensors-23-02468-f001:**
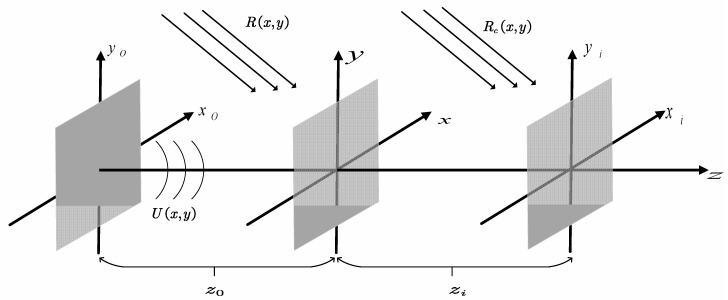
Digital holographic recording and reconstruction.

**Figure 2 sensors-23-02468-f002:**
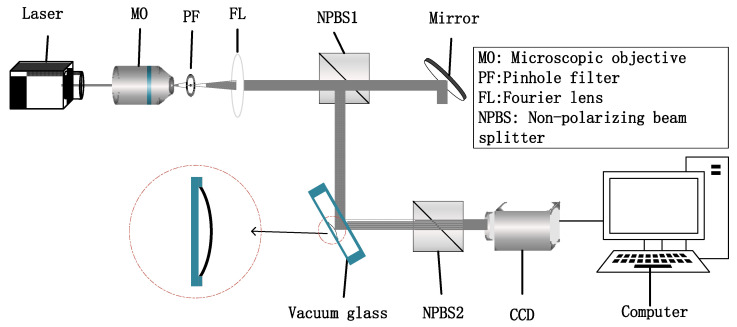
Digital holographic optical path.

**Figure 3 sensors-23-02468-f003:**
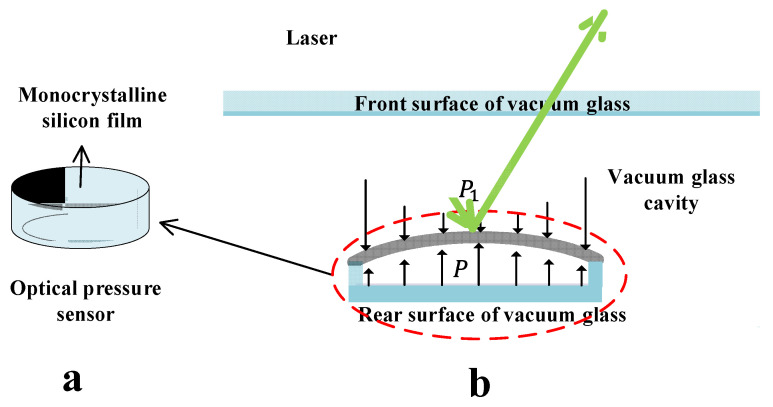
(**a**) Structure diagram of optical pressure light sensor and (**b**) operating principle diagram of optical pressure light sensor.

**Figure 4 sensors-23-02468-f004:**
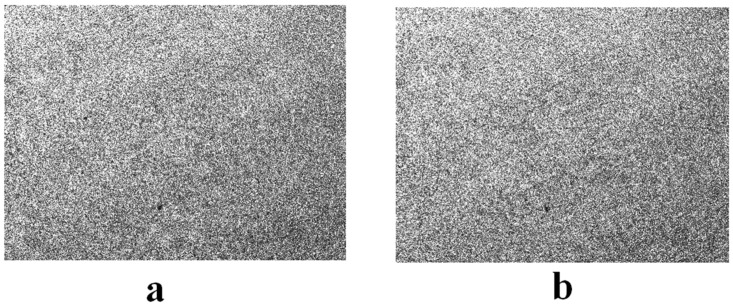
(**a**) Hologram before air leakage and (**b**) hologram after air leakage.

**Figure 5 sensors-23-02468-f005:**
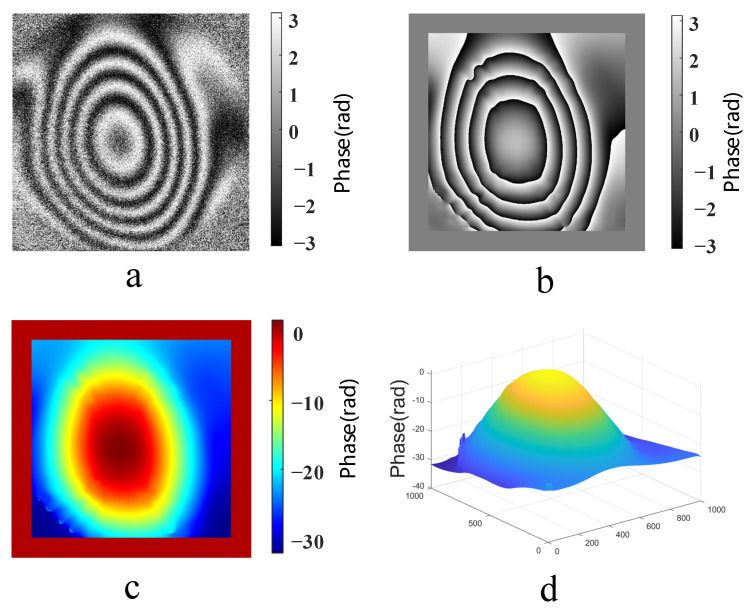
(**a**) Wrapped phase difference map without noise reduction, (**b**) wrapped phase difference map after noise reduction, (**c**) unwrapped phase difference map and (**d**) three-dimensional map of the unwrapped phase difference.

**Figure 6 sensors-23-02468-f006:**
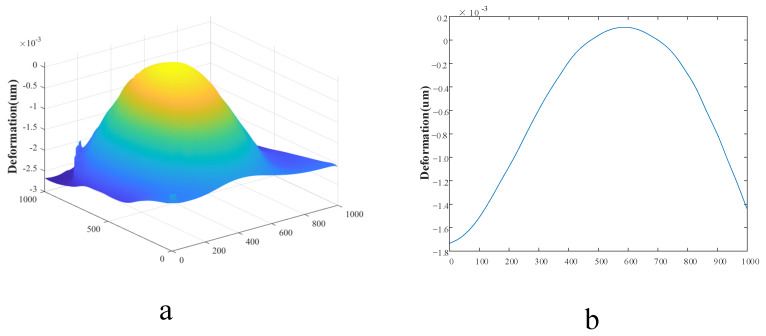
(**a**) Three-dimensional map of deformation and (**b**) the middle profile of Figure 6a.

**Figure 7 sensors-23-02468-f007:**
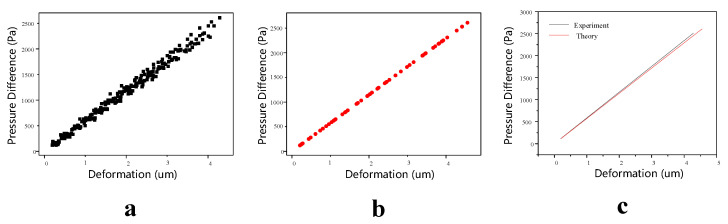
(**a**) Experimental results, (**b**) theoretical results and (**c**) comparison of the two results.

**Figure 8 sensors-23-02468-f008:**
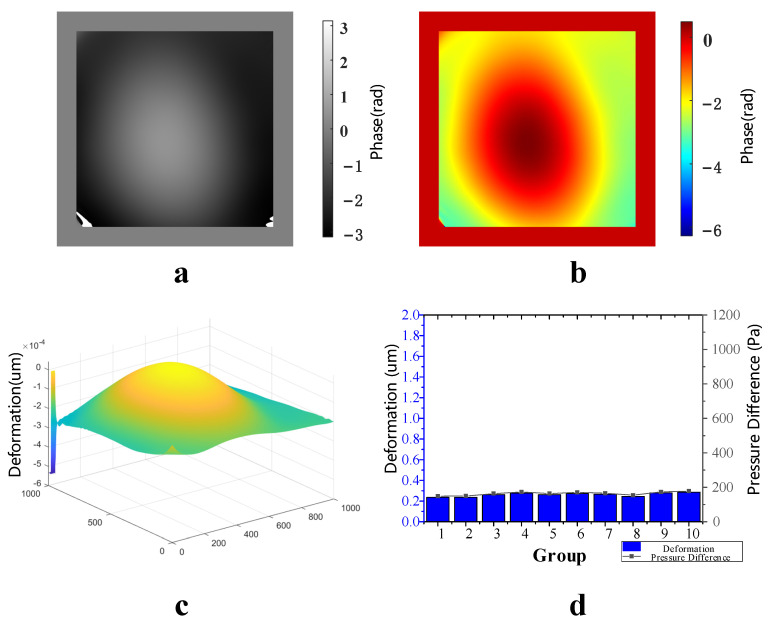
Measurement results with an initial vacuum of −5 kpa. (**a**) Wrapped phase difference map after noise reduction, (**b**) unwrapped phase difference map, (**c**) three-dimensional deformation map and (**d**) 10 deformation measurements and corresponding air leakages.

**Figure 9 sensors-23-02468-f009:**
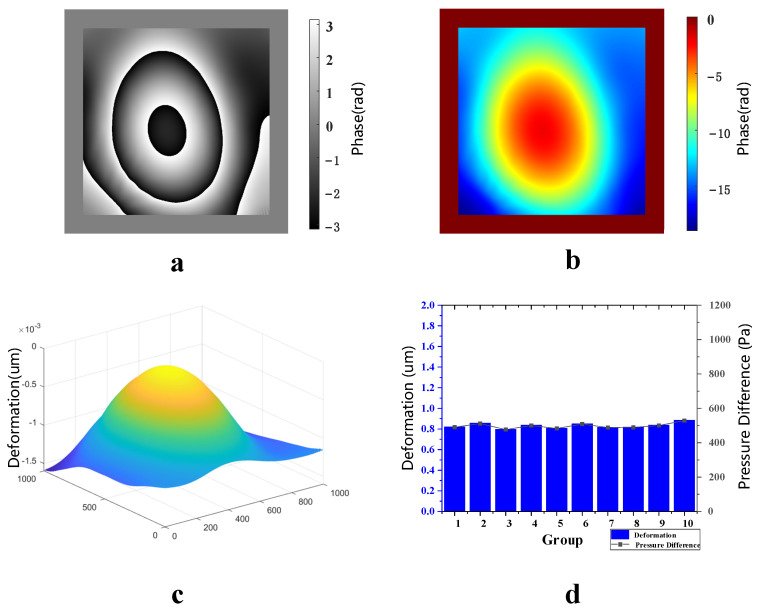
Measurement results with an initial vacuum of −10 kpa. (**a**) Phase difference map of package, (**b**) real phase difference map, (**c**) three-dimensional deformation map and (**d**) 10 deformation measurements and corresponding air leakages.

**Figure 10 sensors-23-02468-f010:**
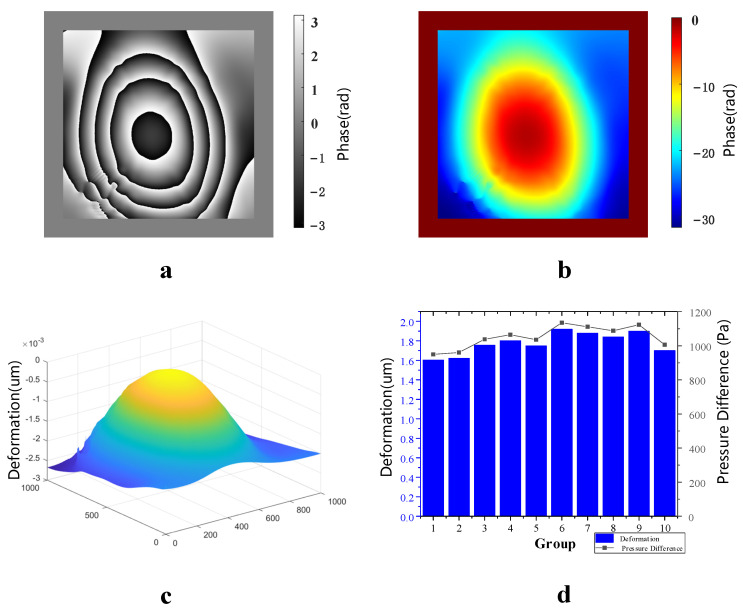
Measurement results with an initial vacuum of −15 kpa. (**a**) Phase difference map of package, (**b**) real phase difference map, (**c**) three-dimensional deformation map and (**d**) 10 deformation measurements and corresponding air leakages.

**Table 1 sensors-23-02468-t001:** Parameters of monocrystalline silicon film.

Diameter/mm	Thickness/mm	Density/(kg/m^3^)	Elastic Modulus/Gpa	Poisson’s Ratio
14	0.1	2330	180	0.2

## Data Availability

Data will be made available on request.

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
