# Peer review of "A Method for Detecting the Vacuum Degree of Vacuum Glass Based on Digital Holography"

_sensors, 2023, doi:10.3390/s23052468_

Round 1
Reviewer 1 Report
The paper "A Method for Detecting Vacuum Degree of Vacuum Glass Based on Digital Holography" shows a sensible measurement setup for printing within a transparent object using the example of vacuum glass, which is of increasing industrial interest.
Holography is used to measure the deformation of a membrane, here silicon, very accurately (sub wavelength). This is not new, but elegantly solved.
The selected Mach-Zehnder arrangement seems to be sensibly chosen for this, since such disturbing reflections on the first glass do not get into the sensor.
The results are presented conclusively and comprehensibly with some limitations:
For the graphics, the scaling (image scale) and a color scale for the color coding or grayscale would have to be added. Please in physical units (not pixels etc.).
Similarly, some diagrams (Figs. 4, 7, 8) lack the physical units altogether.
In Fig. 5, the units are no longer correct if 10 x -3 and 10 x-4 are additionally on the y-axis. I assume that we have a deformation in the range of a few single μm here. This then matches the number of stripes of the reconstructed phase images.
Line 270 and line 287 missing units (standard deviation).
Basically, I would be interested to know whether this method is really applicable in industrial production? For each measurement, a pressure sensor must be placed between the glass plates in vacuum. If the sensor is removed, the vacuum is destroyed. I can imagine working in the development laboratory.
Then the measurements show a very nice reproducibility. This also suggests a high absolute accuracy. Here, however, I suspect a fundamental problem of the method. Can the authors please comment on how the method can be absolutely calibrated and how good the repeatability is?
Author Response
Thank you for your review on our paper. We have answered each of your points below.
- For the graphics, the scaling (image scale) and a color scale for the color coding or grayscale would have to be added. Please in physical units (not pixels etc.).
Based on your suggestions, the relevant content has been modified.
- Similarly, some diagrams (Figs. 4, 7, 8) lack the physical units altogether.
Based on your suggestions, the relevant content has been modified.
- In Fig. 5, the units are no longer correct if 10 x -3 and 10 x-4 are additionally on the y-axis. I assume that we have a deformation in the range of a few single μm here. This then matches the number of stripes of the reconstructed phase images.
Based on your suggestions, the relevant content has been modified.
- Line 270 and line 287 missing units (standard deviation).
Based on your suggestions, the relevant content has been modified.
- Basically, I would be interested to know whether this method is really applicable in industrial production? For each measurement, a pressure sensor must be placed between the glass plates in vacuum. If the sensor is removed, the vacuum is destroyed. I can imagine working in the development laboratory.
This method can be applied to industrial production, because the overall thickness of the sensor can be made in the micron level, and the occupied area can be in the millimeter level. In the experiment, due to the limitation of the laboratory processing capacity, the film produced is relatively large, but the effective working area is in the millimeter level. Compared with vacuum glass, tthe micro-size sensor can be placed in the vacuum glass near the edge , so it will not affect the use performance. When leaving the factory, the sensor is placed inside the vacuum glass, which can monitor the vacuum degree of the vacuum glass in real time during its whole life, and it does not need to be removed during use.
- Then the measurements show a very nice reproducibility. This also suggests a high absolute accuracy. Here, however, I suspect a fundamental problem of the method. Can the authors please comment on how the method can be absolutely calibrated and how good the repeatability is?
This is a very good question. In the experiment, the recording distance of each experiment is consistent, and the error has little influence on the detection results, but there will be a small deviation in the observation direction. We first measure the initial morphology of the film, and make angle compensation according to the 5 * 5 points. The experimental results show that each compensation value is very small, and correct the fitting curve together with the changes in the external environment such as (pressure, temperature) and other conditions, The built-in optical pressure sensor under this condition can be calibrated without obvious impact on the test results. This method has good repeatability.
Thank you very much for your pertinent comments. The questions you put forward are particularly good and have great inspiration for us.
Reviewer 2 Report
Dear Authors,
In my opinion, the paper requires some extensive revision. Please, take into consideration my points from the attached document.

Author Response
Thank you for your review on our paper. We have answered each of your points below.
- 2 and thecorresponding text-this part of the research leaves a lot of questions. What is the nominal or starting shape of this capsule and how is it controlled? How do you control the pressure inside the capsule? And how to be sure that if the big vacuum cell has a leakage, this small volume never has leakages in its turn? How is it made in practice? And why the monocrystalline Si film is used (it seems to be difficult in manufacturing and fragile)?
The shape of capsule is a spherical crown, and its shape parameters are determined by the pressure difference inside and outside the reference cavity. In practical applications, the initial pressure of the reference chamber can be determined according to the material characteristics of the pressure-sensitive film. During the processing, the sealing process of the reference chamber is completed in a controlled negative pressure environment, and the negative pressure environment pressure is the reference chamber pressure. In the manufacturing process of optical pressure sensor, the silicon base material at the bottom is integrated with the pressure sensing film, so the possibility of leakage is low. If the leakage amount of the external chamber is too large, the pressure sensing film will have a large deformation, and the leakage detection of the vacuum glass can still be realized. The monocrystalline silicon film is widely used in the semiconductor industry, with high processing accuracy, multiple suppliers and stable. At the same time, the sensor is placed inside the vacuum glass, which is not easy to be damaged. Within the pressure range described in this paper, 100 μ m monocrystalline silicon film can be well used for vacuum detection. For different application scenarios and detection ranges, other materials with easy access and stable properties can also be used to make pressure-sensitive films.
- 9 and the entire measurement technique-it seems that the method allows to measure only the change of the internal pressure via the membrane deformation. It is unknown, what was the starting curvature of the Si film and the starting vacuum degree. So the method can t provide the absolute value of the internal pressure. In the meantime for the safety control, maintenance or lifecycle prediction and other tasks I would expect the absolute value to be required .
In the current research process, we obtain the vacuum degree of the vacuum glass in service by detecting the film deformation caused by the pressure difference change and combining the initial pressure value of the vacuum glass. The influence of the absolute value of the internal pressure of the vacuum glass on the detection result can be removed by introducing the digital holographic detection method to detect the phase difference. In the follow-up research, we can try to accurately control the initial vacuum degree of the reference cavity, so that we can calibrate the detection system more accurately, and achieve higher precision detection of the absolute value of the internal pressure of the vacuum glass.
- It is not clear how to use this method in practice. The authors claim that it has some commercial use prospective. But it seems like the control membrane must be incorporated into every vacuum glass cell on the manufacturing stage. Would it affect the glass cell properties like transparency. Even if it is possible, how to check the vacuum degree once the item is in use? What would be a device/setup for it and how to calibrate it? Taking into account the issue with the absolute values measurements it may appear that one has to measure the pressure change repeatedly and to compare them with some stored old results.
In our paper, the thickness of the sensor can be made to micron level, and the area occupied by the sensor can be made to millimeter level. At present, this research is still in the laboratory research stage, but it can be seen from the optical path diagram 1 that the optical path of the system is simple and it is easy to make detection instruments. The parameters to be calibrated are mainly detection distance, observation angle and environmental parameters. In the experiment, the recording distance of each experiment is consistent, and the error has little influence on the detection results, but there will be a small deviation in the observation direction. We first measure the initial morphology of the film, and make angle compensation according to the 5 * 5 points. The experimental results show that each compensation value is very small, and correct the fitting curve together with the changes in the external environment such as (pressure, temperature) and other conditions, The built-in optical pressure sensor under this condition can be calibrated without obvious impact on the test results. This method has good repeatability.
- The paper needs some extensive language correction, but more attention should bepaid to use of correct terms:" Mach-Zender optical path " = "Mach-Zender interferometer", "computer program" = "software" , "carbon energy saving" "carbon emission saving" or "energy saving "?, "parallel light" = "parallel beam" , "diffraction section" = "diffraction order" ? and other examples.
We have revised the relevant terms you mentioned and improved the English writing of the whole article.
- Would it be more correct to talk about a vacuum glass cell rather than about just vacuum glass?
Yes, the focus of this paper is to propose an optical detection method and device that can be used for vacuum detection of vacuum glass cavity, so it is modified as vacuum glass cell is more correct.
- Line 77- the equation is derive for a spherical replaying wavefront, but in the experiment only flat wavefronts seem to be used (it is a particular case of a spherical wave R= , but still).
As you said, plane wave is a special kind of spherical wave. In the previous literature, it is more common to derive the relevant formula using spherical wave as the reconstruction light wave, so this expression is used in the article.
- Also, the theoretical background starts with the holography equation, while in the real measurements the hologram reconstruction is neither performed physically, nor simulated in So it is an open question if this technique belongs to digital holography or to interferometry in general.
This method needs to calculate the deformation of the film through the phase difference, so as to realize the detection of vacuum degree. Therefore, it needs to accurately reconstruct the object light field through digital hologram to realize the detection of phase difference. The system must introduce the detection method of digital holography, which belongs to digital holographic interferometry.
- Subsection 2.1- perhaps it would be useful to have a drawing to comment the main variables and compare the theory and the experimental setup.
According to your suggestion, we have added Figure 1 to represent the main variables and explained the relationship between the theory and the experimental device.
- 1- why the collimating lens is indicated as a spatial filter? In a system with laser source I would expect to have a focusing lens, a pinhole (which is actually a spatial filter) and a collimating lens. This is a standard way to get a collimated beam of a necessary diameter and rid out of specles.
The spatial filter in this paper includes microscopic objective and pinhole, which play the role of beam expansion and spatial filter respectively. Based on your suggestion, we have modified it.
- Line 151-how the temperature was stabilized and controlled?
The experiment was conducted in a constant temperature room, and the temperature difference before and after the test was less than ± 0.5 ℃.
- 4and other experimental results what are the recording wavelength and the source power?
The recording wavelength is 532 nm, and the source power is 200 mw.
- Line 198-83 microns is not a subwavelength value unless it is a CO2 laser or other IR source.
The description of subwavelength has been modified, and the subwavelength has been changed to the negative quadratic level of 10.
- Fig. 7,8 and 9 and the corresponding text. The results and descriptions are almost identical,sothey could be presented shorter. Also, the standard deviation is exactly the same in all the cases, which is very suspicious, and the units are not given.
Based on your suggestion, we have simplified the corresponding content. Due to the negligence of submitting the manuscript and the error of standard deviation, we have updated it.
Thank you very much for your pertinent comments. The questions you have raised are particularly good and have great inspiration for us.

Round 2
Reviewer 2 Report
Dear Authors,
Thank you for taking my remarks into account.
However, I have a couple of additional points:
1. It seems that the statement about the capsule shape relies on some approximation. I would expect this surfae to be rather a general 2nd order axisymmetric surface (or even more complex shape if the contact ring is wide enough) than exactly a sphere. Perhaps, some more theoretical computation is needed to show that the shape is close enough to sphere(e.g. with the theory from P. Seide Small elastic deformations of thin shells Springer, 1975). Alternatively, it is possible to show that this potential deviation from a sphere is negligable and/or provide some references to support the presumption.
2. I'm missing a bit the context regarding the vacuum glass cell manufacturing and use: What is an accaptable change of the vacuum level and which change would be dangerous? And ho these values comparee with the obtqained measurement range and sensitivity? How long one cell can be in use and is it mandatory to re-verify the pressure? Would your setup be targeted on some stages of manufacturing or rather on the entire lifetime of the cells? Having the answers with some references could help to understand the industrial pospects better. At the moment it seems that the testing approach is slightly ovecomplicated.
Author Response
Thank you very much for your question, which points the way for our next in-depth study. I have responsed to these questions point by point as follows,
- It seems that the statement about the capsule shape relies on some approximation. I would expect this surfae to be rather a general 2nd order axisymmetric surface (or even more complex shape if the contact ring is wide enough) than exactly a sphere. Perhaps, some more theoretical computation is needed to show that the shape is close enough to sphere(e.g. with the theory from P. Seide Small elastic deformations of thin shells Springer, 1975). Alternatively, it is possible to show that this potential deviation from a sphere is negligable and/or provide some references to support the presumption.
As you said, the deformation of thin film follows the small deflection principle of thin shells. We have added relevant references( [26, 27] ). However, in the detection method described in this paper, the shape of the film has little effect on the results. We adopted the cross-subtraction method of pressure difference to eliminate the deviation caused by different initial morphology of the film, and the mean value of multiple deformations at and around the maximum deformation as the final measured value to calibrate the deformation and pressure difference in the full range (0-2600pa).
- I'm missing a bit the context regarding the vacuum glass cell manufacturing and use: What is an accaptable change of the vacuum level and which change would be dangerous? And ho these values comparee with the obtqained measurement range and sensitivity? How long one cell can be in use and is it mandatory to re-verify the pressure? Would your setup be targeted on some stages of manufacturing or rather on the entire lifetime of the cells? Having the answers with some references could help to understand the industrial pospects better. At the moment it seems that the testing approach is slightly ovecomplicated.
The vacuum detection range of the optical pressure sensor described in this paper is 0-2600 Pa. If the material and size of the film and the vacuum degree of the cavity in the pressure sensor change, the corresponding vacuum detection range and application range will also be different. With this feature, we can optimize the design for different application scenarios. In pressure detection, if there is a large amount of air leakage in the sensor cavity, the sensor's measurement of vacuum will fail.Under normal conditions, the sensor will not have serious leakage. If the pressure change caused by leakage is far less than the sensitivity and detection range of the digital holographic detection system, the sensor can be used all the time, so it is not necessary to re-verify the pressure. In industrial manufacturing, the pressure of the two cavities should be set very close at the beginning. The probability of leakage of the large cavity is much greater than that of the small cavity. Once the large cavity leaks, the sensor will produce corresponding deformation.The sensor we designed can be used for monitoring the whole life of glass in use. And the digital holographic detection system can be made into a separate detection system, and the accessories we use are universal. We have added relevant references( [14], [33] ) which help us understand the industrial pospects better.
